# The Latest Prevalence, Isolation, and Molecular Characteristics of Feline Herpesvirus Type 1 in Yanji City, China

**DOI:** 10.3390/vetsci11090417

**Published:** 2024-09-07

**Authors:** Meng Yang, Biying Mu, Haoyuan Ma, Haowen Xue, Yanhao Song, Kunru Zhu, Jingrui Hao, Dan Liu, Weijian Li, Yaning Zhang, Xu Gao

**Affiliations:** Laboratory for Animal Molecular Virology, Department of Veterinary Medicine, College of Agricultural, Yanbian University, Yanji 133002, China; y2439082877@outlook.com (M.Y.); mubiying@sina.com (B.M.); bakougenn@outlook.com (H.M.); xhw704601416@foxmail.com (H.X.); syh20230516@163.com (Y.S.); zkr13596766195@163.com (K.Z.); m17808001838@163.com (J.H.); ld1952415537@163.com (D.L.); lwj13180618872@163.com (W.L.); zyn55082023@163.com (Y.Z.)

**Keywords:** gD gene, mutation, sequence analysis

## Abstract

**Simple Summary:**

Viral rhinotracheitis is an acute upper respiratory (URT) disease caused by feline herpesvirus type 1 (FHV-1). Cats remain in a latent infectious condition after healing from the primary infection. Therefore, there is no viable treatment for FHV-1 infection. This disease is widely distributed worldwide and poses a threat to the health of pet cats and the protection of rare felines. To further understand the current epidemic trend of FHV-1, we isolated the virus from Yanji City for the first time, revealing the genetic evolution direction of FHV-1 by comparing the gD gene sequences between 33 positive samples and those uploaded worldwide. This study provides a theoretical foundation for the prevention and treatment of feline viral bronchitis by conducting an epidemiological investigation.

**Abstract:**

Epidemiological surveys revealed that 33 of the 93 samples were positive for FHV-1, with the gD gene of these 33 samples exhibiting low variation, high homology, and no critical amino acid mutation. Feline herpesvirus type 1 (FHV-1), also known as feline viral rhinotracheitis (FVR) virus, is one of the main causes of URT disease in cats. All cats can become hosts of FHV-1, and the spread of this disease affects the protection of rare feline animals. Nasal swabs from cats with URT disease were collected at five veterinary clinics in Yanji City from 2022 to 2024. The purpose of this study was to isolate and investigate the epidemiology of FHV-1. The gD gene of the FHV-1 strain was cloned and inserted into the pMD-18T vector and transformed into a competent *Escherichia coli* strain. Subsequently, the gD gene of the positive samples was sequenced and phylogenetic analysis was performed to determine the genetic evolution relationship between the strains. We successfully isolated the FHV-1 strain YBYJ-1 in Yanji City for the first time. The diameter of the virus is approximately 150–160 nm. After 48 h of virus inoculation, the cells were round, isolated, and formed grape-like clusters. The gD gene of the virus was sequenced, and the length was 1125 bp, which proved the isolate was FHV-1. This study found that the genetic evolution of the FHV-1 gD gene was stable, expanding the molecular epidemiological data on FHV-1 in cats in Yanji City.

## 1. Introduction

Feline herpesvirus is a member of the *Herpesviridae* family, alphaherpesvirus subfamily, and the varicellovirus genus. There is only one serotype of the virus, and it is genetically relatively homogenous [1]. In 1958, Crandell and Maurer first reported the presence of FHV-1 in cats with respiratory diseases [2], and the virus was subsequently isolated and identified worldwide. The most common diseases caused by FHV-1 in the clinic are respiratory and eye diseases [1]. Infected cats will have conjunctivitis, corneal ulcers, dyspnea, fever, depression, and other symptoms [3]. In severe cases, secondary bacterial infection and fibrin-necrotizing bronchopneumonia may lead to death in cats [4]. After FHV-1 infection in cats, the lesions mainly occur in the upper respiratory tract [5,6]. Diffuse hyperemia and ulcers can occur in the turbinate bone of the cat, and tissue congestion can occur in the cervical lymph nodes [7]. Once infected with FHV-1, the host is a lifelong carrier of the virus, which seriously affects its health [8]. Young cats are more susceptible to primary infection than adult cats and have a greater mortality rate [9].

Feline herpesvirus is an enveloped DNA virus with a linear double-stranded DNA genome length of 126–134 kb [10]. The G + C content is approximately 50%. It consists of a unique long region (UL) of approximately 99 kb and a unique short region (US) of approximately 27 kb. On both sides of the US are a pair of identical inverted repeats, called terminal short repeats (TRs) and inverted short repeats (IRs) [11]. FHV-1 contains 78 open reading frames (ORFs), encoding seventy-four different proteins, including twenty-three nucleocapsid-associated proteins [12] and thirteen glycoproteins in the virus envelope, of which eight glycoproteins have been identified. They are gB, gC, gD, gE, gG, gH, gI, and gL [13]. The glycoprotein D is relatively conserved throughout the α-herpes subfamily, with only minor changes. The US region of the FHV-1 genome contains a 6.2 kb nucleotide sequence; this sequence contains five ORFs, and the third ORF encodes glycoprotein D, which is the only glycoprotein encoded by a gene from the US region of FHV-1 that is essential for virus production. The glycoprotein D can specifically bind to cell surface molecules and plays an important role in virus penetration into vulnerable cells [14]. Therefore, glycoprotein D has the potential to develop efficient subunit vaccines. In addition, FHV-1 gD may be one of the factors that determines the host range of FHV-1 [13].

FHV-1 has a wide range of transmission routes and is highly infectious. It can be transmitted directly by contact with secretions or indirectly by contact with contaminated objects. The virus can easily spread from respiratory and ocular secretions to susceptible cats. FHV-1 is latent in the trigeminal ganglia in virtually all infected but clinically recovered cats [15], and in this sense, it is difficult to eradicate the virus on a population basis. For the health of cats, the first dose of vaccine should be inoculated at the age of 8 weeks, the second dose should be inoculated after 3–4 weeks, and the third dose should be inoculated after 3–4 weeks. However, vaccination can not completely prevent the spread of the virus [16], and there are individual differences in the protective effect of the vaccine on cats [17]. Therefore, a combination of vaccination and management control should be adopted.

Many cases of FHV-1 infection in cats have been reported worldwide [18,19,20,21], and FHV-1 has also been detected and isolated in Beijing, Shanghai, and other places in China [22]. Yanji City is located at the border of China, North Korea, and Russia. It has 13 nature reserves and rich wildlife resources, such as wild leopards and tigers, which need targeted protection. In recent years, there have been frequent exchanges with foreign cat breeds, and the protective effect of the vaccine is limited. At present, there is no investigation or research on FHV-1 in Yanji City. Therefore, the main purpose of this study was to investigate the epidemic status of FHV-1 and to provide theoretical guidance for the prevention and treatment of FHV-1 in Yanji City by isolating local FHV-1 strains, as well as for the further development of vaccines and targeted drugs.

## 2. Materials and Methods

### 2.1. Clinical Samples and Cells

Nasal swab samples of 93 cats with URT disease were collected at an Animal Hospital for follow-up epidemiological investigation in Yanji City from 2022–2024. PCR results showed that 33 of the 93 samples were positive, and one positive sample was randomly selected for virus isolation and stored in a −80 °C refrigerator for subsequent research by our team. The sample was initially screened by PCR, and only FHV-1 tested positive, with no feline parvovirus (FPV), feline calicivirus (FCV), or feline coronavirus (FCoV) detected. The samples were stored in a −80 °C refrigerator at the preventive veterinary laboratory of the Agricultural College of Yanbian University until analysis. The F81 cell line was provided by the China National Center for Veterinary Culture Collection.

### 2.2. Isolation and Identification of Viruses

Nasal swabs of suspected FHV-1-infected cats with symptoms such as conjunctivitis, dispirited, stomatitis, and sneezing were collected and stored in 0.9% normal saline at 4 °C. The centrifuge tube that contained the swab sample was then transported to the laboratory within 1 h, and the liquid was centrifuged at 5000 rpm for 10 min. The supernatant was filtered using a 0.22 μm filter membrane (Biosharp, Shanghai, China) and then inoculated into F81 cells with a culture density of 80%. The F81 cell is an engineering cell line screened by primary cells of a cat kidney through subculture cloning. Feline herpesvirus is relatively easy to replicate within it. The culture conditions are DMEM + 10% FBS + 1% P/S (Gibco, Walsham, USA). The cells were cultured in a 37 °C with 5% CO_2_ incubator_._ Cell changes were monitored daily until cytopathic alterations (CPE) exceeded 80% of the total cell volume. The collected virus suspension was repeatedly frozen and thawed three times, followed by centrifugation at 5000 rpm for 5 min. Then, the supernatant was harvested and transferred to a fresh F81 culture flask. The cells were passaged for six passages under the same culture conditions. Concurrently, F81 cells not exposed to the virus were used as the blank control group.

According to the full-length sequence of the FHV-1-gD gene (accession number: MT813047) published in the GenBank database, Oligo7.0 (Molecular Biology Insights, Amsterdam, USA) was used to design primers for amplifying the entire gD gene sequence (gD-F: 5′-ATGATGACGTCTACATTTTTG-3′; gD-R: 5′-TTAAGGATGGTGAGTTGTATGTAT-3′). The PCR total reaction volume was 25 μL, which included 12.5 μL of 2 × Premix Taq (Takara, Tokyo, Japan), 2 μL of primer pairs, 1 μL of DNA sample, and 9.5 μL of ddH_2_O. The PCR amplification process was as follows: predenaturation at 94 °C for 5 min, 35 cycles of denaturation at 94 °C for 30 s, annealing at 56 °C for 1 min, and extension at 72 °C for 1 min, with a final extension for 10 min at 72 °C. The expected length of the gD gene is 1125 bp.

### 2.3. Transmission Electron Microscopy (TEM)

The 6th generation of FHV-1 virus solution was concentrated using a hollow fiber column (Rigorous, Shenzhen, China), and then copper mesh penetration (GE Healthcare, Marlborough, MA, USA), glutaraldehyde fixation, and 2% phosphotungstic acid staining. Subsequently, the morphology of the virus particles was observed using a transmission electron microscope (Hitachi, Tokyo, Japan), and high-resolution images were taken using a digital camera (Olympus, Tokyo, Japan). This experiment was performed in a sterile laboratory.

### 2.4. Indirect Immunofluorescence Identification

The virus suspension was injected into the culture flask when the cell growth density reached 80%, and a culture flask without virus inoculation was used as the control group. After 48 h of culture, 4% paraformaldehyde (Biosharp, Shanghai, China) was added to fix the cells once cytopathic effects were observed. Triton X-100 (Solarbio, Beijing, China) was added to make the cell membrane transparent. Following blocking with 5% skim milk at 37 °C, a mouse anti-FHV-1 primary antibody (Kejie, Shenzhen, China) was added and incubated overnight. The next day, the FITC-labeled goat anti-mouse IgG antibody (Abcam, Cambridge, UK) was incubated at 37 °C in the dark. The cells were observed under an inverted fluorescence microscope.

### 2.5. T-A Cloning and Molecular Epidemiology of FHV-1

The gel recovery kit (Omega Bio-tek, Norcross, GA, USA) was used to recover the PCR product, and the purified product was ligated to the pMD-18T vector (Takara, Tokyo, Japan) and transformed into competent Trans5α (TransGen Biotech, Beijing, China) cells to obtain the recombinant plasmid pMD18T-FHV-1, which was subsequently sent to Changchun Kumei Biotechnology Company for sequencing. A total of 93 clinical samples were amplified by using the above PCR detection program, and the positive samples were sequenced. The gD gene sequences were deposited in GenBank.

### 2.6. DNA Sequencing and Phylogenetic Analysis

The gD gene sequence was cut into fragments with a length of 1125 bp by EditSeq in DNAStar (DNASTAR, Inc., Madison, WI, USA). The sequencing results were compared with the FHV-1-gD gene published in the NCBI GenBank database (http://www.ncbi.nlm.nih.gov, accessed on 20 July 2024). The Mega 11 (Mega Limited, Auckland, New Zealand) was used for phylogenetic analysis and sequence alignment for the amino acid sequences of the gD gene obtained from sequencing. The neighbor-joining method was used to construct the gD gene phylogenetic tree, and the reliability was analyzed using the bootstrap test with 1000 replications. The DataMonkey website (https://www.datamonkey.org/, accessed on 22 July 2024) was used to test the selection pressure of the YBYJ-1 strain and other reference strains uploaded to the NCBI GenBank database. The FUBAR method was used to infer the likelihood of the evolutionary direction of the YBYJ-1 strains, which was based on a positive selection pressure higher than 0.9. (A posterior probability > 0.9 strongly suggests positive selection).

## 3. Results

### 3.1. Preliminary Identification of Clinical Samples

The viral DNA was amplified using PCR, and 33 of 93 suspected samples were detected. The positive detection rate was 32.2%. Agarose gel electrophoresis revealed a single DNA band at approximately 1200 bp (Figure 1), which corresponded to the expected size of the gD gene (1125 bp), while the PCR amplification of the feline parvovirus (FPV), feline calicivirus (FCV), and feline coronavirus (FCoV) samples showed no target bands. In this study, the initial agarose gel data have been uploaded as Appendix A.

### 3.2. Data Processing of Positive Samples

In this molecular epidemiological survey of FHV-1 in Yanji City, it was found that cats under 6 months were more likely to be infected with FHV-1, but no association between breed and FHV-1 infection was found. The proportion of the samples identified as FHV-1 positive is shown in the relevant sections (Table 1) below. The gD gene sequences of 33 positive samples identified by PCR amplification have been uploaded to the GenBank database (accession numbers: OQ587981, OQ686817-OQ686828, OQ689760-OQ689761, andOQ710115-OQ710132).

### 3.3. Sequence Analysis of the gD Gene

Among the 33 FHV-1 strains in this study, 1/33 (3%) had the Tyr71Cys, Asn172Asp, and Ala265Thr mutations, and 3/33 (9%) had the Leu16Pro, Lys17Glu, Val72Ala, Pro241Ser, Asp279Gly, and Met305Ser mutations. The results showed that the gD gene was relatively conservative and had fewer mutations. The homology analysis of 33 sequences in this study and 30 sequences uploaded from the GenBank database revealed that the amino acid homology of the 33 FHV-1 gD sequences and 30 reference strains remained between 98.9 and 100%. The results showed that the 33 FHV-1 sequences had high homology with those of the 30 reference strains. The selection pressure analysis of the 63 FHV-1 sequences mentioned above showed that only the 241st amino acid of the gD gene was under positive selection, and the Bayes factor was 15.730, indicating that the gD gene is relatively conserved.

### 3.4. Phylogenetic Analysis

Phylogenetic analysis showed that the 33 FHV-1 strains involved in this study were closely related to the reference strains in China and worldwide, which may have evolved from the same or similar ancestral mutations (Figure 2).

We also selected other host herpesviruses, including equine herpesviruses (EHV-1) OP271695 and KF644579; bovine herpesviruses (BHV-1) OP035381 and AJ004801; herpes simplex viruses (HSV-1) ON023028 and ON513441; pseudorabies viruses (PRV) ON005002 and OP376823; and canine herpesviruses (CHV-1) AF361076 and KT819633, for a total of 10 reference strains. The phylogenetic analysis of all sequences using Mega 11 showed that 33 FHV-1 sequences and 10 other host herpes viruses have a distant relationship (Figure 3).

### 3.5. Virus Isolation

After 48 h, the cells inoculated with FHV-1 exhibited obvious cytopathic effects. Under the inverted microscope, the cells became round, detached, and formed grape-like clusters. The F81 cells in the control group did not exhibit abnormalities (Figure 4).

### 3.6. Transmission Electron Microscopy

The concentrated virus solution of the sixth generation was placed under a transmission electron microscope to observe the virus particles. The diameter of the spherical virus particles was about 150–160 nm, and the morphology was complete (Figure 5).

### 3.7. Indirect Immunofluorescence Identification

The FHV-1-YBYJ1 identified by PCR was inoculated into F81 cells, and the green fluorescence was observed in the inoculated group but not in the control group, indicating that FHV-1 replicated in F81 cells (Figure 6).

## 4. Discussion

This is the first epidemiological investigation of feline herpesvirus in Yanji City, Jilin Province, China. The main aim of this study was to identify the prevalence and variation of FHV-1. Feline herpesvirus infection is often accompanied by fever, salivation, conjunctivitis, ocular and nasal discharges, sneezing, and sometimes coughing, with a poor prognosis [23]. The rare clinical symptoms caused by FHV-1 include pneumonia [24], skin ulcers [25], gingivostomatitis [26], gastritis, and pancreatitis [27]. At present, there is no effective treatment for this disease. Although domestic cats are widely vaccinated, FHV-1 infection is still common [28,29]. Approximately 50–75% of respiratory infections are estimated to be associated with FHV-1. FHV-1 is relatively fragile in the external environment and has no known host or alternative host other than cats [15]. Like other alphaherpesvirus infections, almost all clinically recovered cats are carriers, and latent infection may affect the protective effect of vaccine. Therefore, it is necessary to understand the epidemiological data to develop preventive measures and vaccines. In this study, 93 nasal swabs of FHV-1 collected from cats were analyzed. These cats have clinical symptoms related to the eyes and respiratory systems. The detection rate of FHV-1 in Yanji City was higher than that reported in other countries and regions.

In recent years, an epidemiological survey of the upper respiratory tract of cats in Canada showed that the detection rate of FHV-1 was 29.3% [30]. The epidemiological evaluation of the upper respiratory tract of cats with FHV-1 in Spain revealed that the detection rate of FHV-1 was 28.3% [31]. A retrospective molecular study of feline viral diseases in southern Italy (Campania) showed that the detection rate of FHV-1 was 9.05% [32]. The molecular epidemiological report of a natural infection of feline herpesvirus in Kunshan, southern China, showed that the detection rate of FHV-1 was 21.5% [33]. In the molecular epidemiological investigation of FHV-1 in 16 cities in China, the results showed that the detection rate of FHV-1 was 16.3%, mainly in winter and spring, and the positive rate of FHV-1 in northern China was higher than that in southern cats [34]. In this study, the Yanbian area was in the north of China, and the positive detection rate of the FHV-1 epidemiological investigation was 32.2%, with an increasing infection rate compared to other regions. This indicates the high prevalence of FHV-1 in the Yanbian region and the possible lack of efficacy of the FHV-1 vaccine, reminding us of the urgency of developing an effective vaccine.

In this study, the phylogenetic analysis and selection pressure analysis of gD genes with 30 other reference sequences showed that the genetic relationship between strains was close, the variation was small, and there were mutations at individual sites. This is probably the result of natural selection. FHV-1 gradually adapts to the local natural environment and the internal environment of animal hosts. This study revealed that the FHV-1 gD gene underwent diversified selection at the amino acid 241 site [35]. An epidemiological survey of FHV-1 in Yanji City revealed that most patients with FHV-1 infection were incompletely vaccinated and the time of infection was mostly within 6 months. This finding indicated that FHV-1 positivity was related to age, confirming previous studies [36,37]. Tran et al. reported that a greater proportion of non-pedigree cats had respiratory disease infections [38]. However, in this study, no association was found between FHV-1 infection and the breed of cats. Although several cats from the sample source were vaccinated, they were still infected with FHV-1. The infection may be related to the differences in the breeding environment of multiple cats, the health status and living habits of the cats themselves, and the decrease in maternal antibodies in the kitten before vaccination [39].

With the development of society and improvements in living standards, the rate of pet raising has increased significantly, which has increased the probability of virus transmission. Yanji City is located at the border of China and has many nature reserves. There are many wild tigers, and more targeted protection is needed. Therefore, the local pet cat industry and the protection of rare cats need to carry out molecular epidemiological investigations, isolate local FHV-1 strains, and develop targeted vaccines to prevent the spread of FHV-1.

## 5. Conclusions

The epidemiological survey in Yanji City showed that 33 of the 93 samples were positive, with a positive rate of 32.2%, and in this study, an FHV-1 strain was successfully isolated, named YBYJ-1. PCR amplification revealed that the genomes of 33 FHV-1 strains were highly homologous, with a small variance in the gD gene. The prevalence of FHV-1 described in this study can serve as an initial guide for vaccine development.

## Figures and Tables

**Figure 1 vetsci-11-00417-f001:**
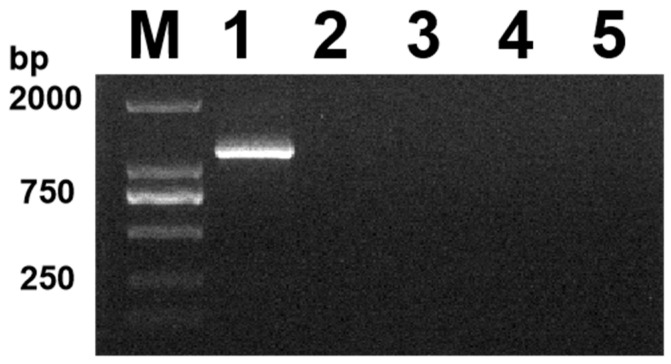
Identification results of PCR, lane M, DL 2000 Marker; lanes 1, amplified products of the isolates; lanes 2–5: FPV, FCV, FCoV, and negative control (ddH_2_O).

**Figure 2 vetsci-11-00417-f002:**
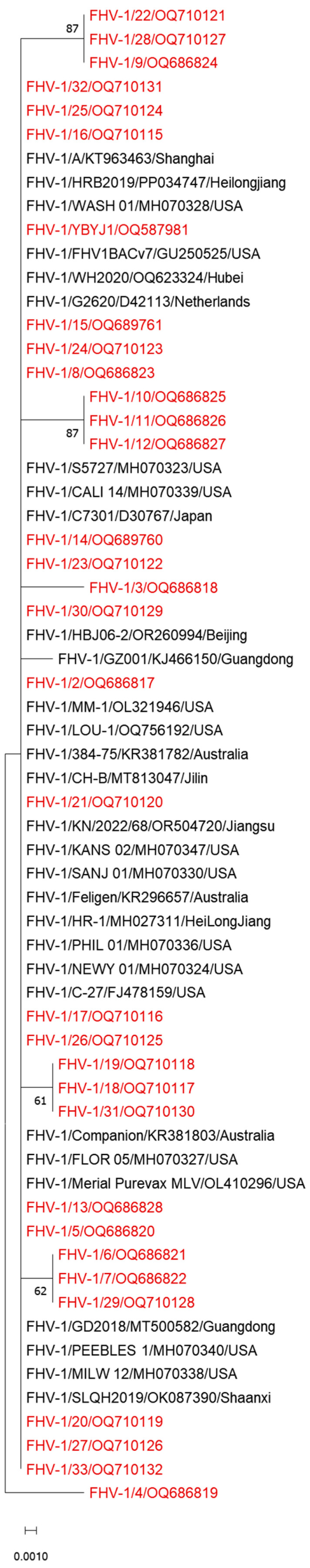
A phylogenetic tree based on the gD sequences of FHV-1 was constructed via the neighbor-joining method. The labels used to distinguish the different strains are as follows: the red label represents the FHV-1 samples involved in this study.

**Figure 3 vetsci-11-00417-f003:**
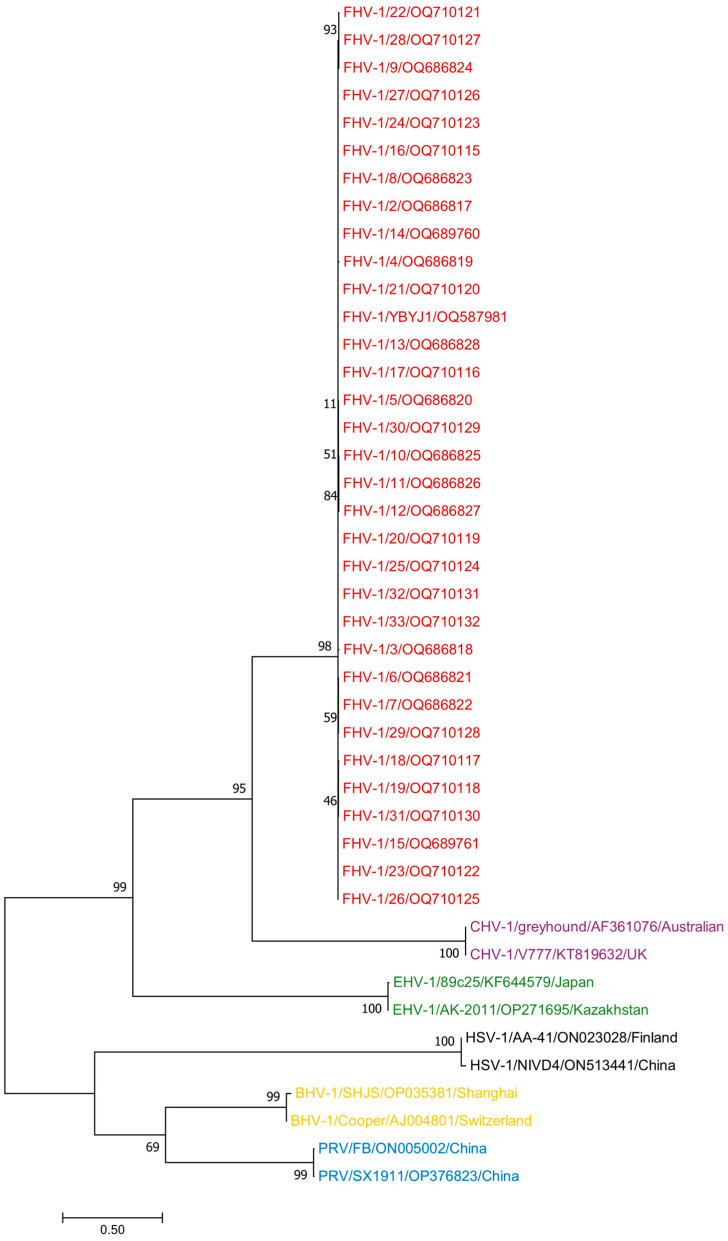
A phylogenetic tree based on the complete gD sequences of FHV-1 and those of EHV-1, BHV-1, HSV-1, and CHV-1 was constructed via the neighbor-joining method. The red label represents FHV-1. The purple label represents CHV-1. The green label represents EHV-1. The black label represents HSV-1. The yellow label represents BHV-1. The blue label represents PRV.

**Figure 4 vetsci-11-00417-f004:**
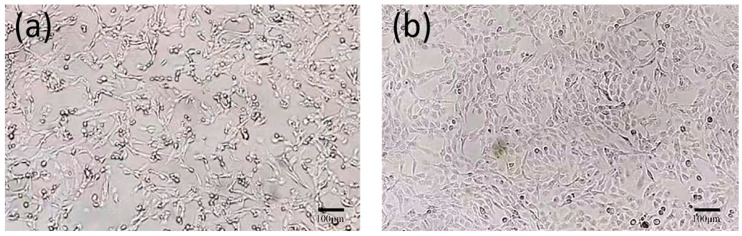
Microscopic observation of cell results (×100). (**a**): Virus-inoculated F81 cells; (**b**): Normal F81 cells.

**Figure 5 vetsci-11-00417-f005:**
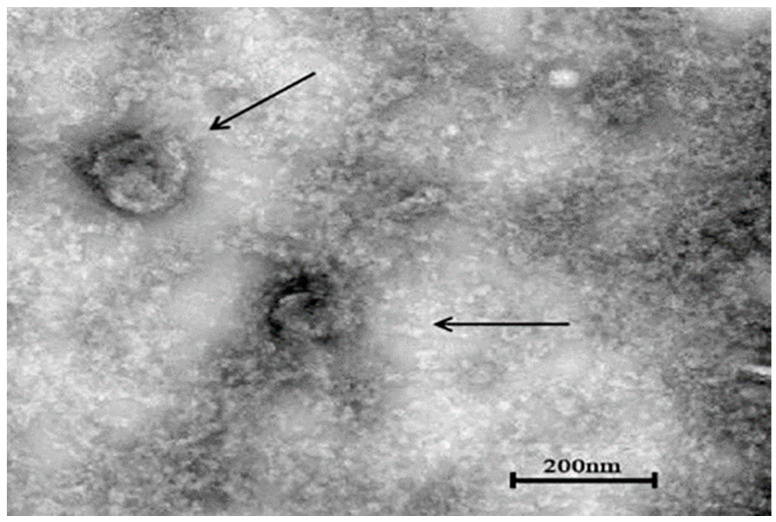
Electron microscope observation results of YBYJ1 (×20,000).cation). Viral particles are highlighted with black arrows.

**Figure 6 vetsci-11-00417-f006:**
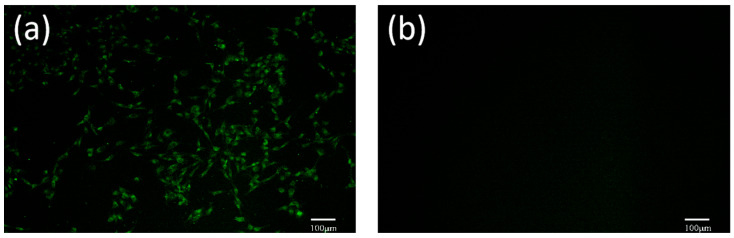
F81 cells infected with the FHV-1-YBYJ1 were detected by indirect immunofluorescence assay (×100). (**a**): Cells infected with the FHV-1-YBYJ1; (**b**): Mock infection control.

**Table 1 vetsci-11-00417-t001:** All FHV-1 positive samples were detected in this epidemiological survey.

Name	Breed	Age (Month)	Clinical Symptoms	Vaccine History	GenBank
1	Maine	4	Snot, sneeze	-	OQ587981
2	American Shorthair	10	Eye conjunctivitis	○	OQ686817
3	British short-haired cat	6	Conjunctivitis, sneezing	-	OQ686818
4	Orange cat	3	Dispirited, fever	-	OQ686819
5	British short-haired cat	8	Fever, eye secretions	-	OQ686820
6	Ragdoll	2	Eye and nose secretions	-	OQ686821
7	Ragdoll	2	Conjunctivitis, sneezing	-	OQ686822
8	British short-haired cat	10	Sneezing	+	OQ686823
9	Chinese Li Hua	2	Conjunctivitis, ocular secretions	-	OQ686824
10	Chinese Li Hua	5	Cough, conjunctivitis	-	OQ686825
11	Orange cat	2	Cough, fever	○	OQ686826
12	American Shorthair	7	Conjunctivitis	-	OQ686827
13	Pastoral cat	9	Dispirited, sneezing	+	OQ686828
14	Siamese	12	Conjunctivitis, ocular secretions	○	OQ689760
15	British short-haired cat	5	Eye and nose secretions	-	OQ689761
16	Garfield	5	Conjunctivitis	-	OQ710115
17	Maine	6	Conjunctivitis, sneezing	○	OQ710116
18	British short-haired cat	8	Sneezing	-	OQ710117
19	British short-haired cat	3	Eye rot bone	+	OQ710118
20	Pastoral cat	3	Conjunctivitis, ocular secretions	-	OQ710119
21	American Shorthair	4	Fever, eye and nose secretions	-	OQ710120
22	American Shorthair	7	Eye and nose secretions, sneezing	-	OQ710121
23	Maine	6	Conjunctivitis, sneezing	○	OQ710122
24	Orange cat	8	Sneezing	+	OQ710123
25	Garfield	12	Conjunctivitis, ocular secretions	○	OQ710124
26	British short-haired cat	2	Eye and nose secretions, Dispirited	-	OQ710125
27	American Shorthair	5	Fever, sneezing and coughing	-	OQ710126
28	Sphynx	2	Fever	-	OQ710127
29	Chinese Li Hua	2	Fever, sneezing	-	OQ710128
30	Ragdoll	3	Conjunctivitis	○	OQ710129
31	Maine	5	Fever, eye and nose secretions	-	OQ710130
32	British short-haired cat	7	Conjunctivitis, ocular secretions	○	OQ710131
33	American Shorthair	4	Fever, eye discharge	-	OQ710132
Prevalence rate = (number of positive samples/total number of subjects) × 100% = 32.2%					

-, unvaccinated vaccine; +, vaccines have been inoculated; ○, incomplete inoculation.

## Data Availability

The data that supports the findings of this study are available in the Appendix Aof this article.

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
