# Peer review of "The Latest Prevalence, Isolation, and Molecular Characteristics of Feline Herpesvirus Type 1 in Yanji City, China"

_vetsci, 2024, doi:10.3390/vetsci11090417_

Round 1

Reviewer 1 Report

Comments and Suggestions for Authors

Yang et al., desribed the detection and isolation feline herpesvirus 1 from feline specimen in a town in north-east China. They prove the specificity of the isolate by PCR and several other methods, gD of the one isolate was sequenced, and were phylogenetically tested.

The work is not too complex, or difficult, but the description of the whole experimental design, and the results is a confusion. Contents of Math and meth. and results are scattered by sentence to semtence at several places in the text. Math and meth and beginning of the result should be rewritten. What kind of samples they used, where from, etc What is the applied cell line, and culturing conditions are missing. Similarly relatively large number of small spelling mistakes were found throughout the paper. FHV-1 is a frequent respiratory viral pathogen of cats. What would the authors give to our knowledge by detection, isolation of an other FHV-1 strain from cats? By finding the same prevalence (around 30%) as other researchers have already found? Why did they do the whole work?

Abstract

line 16 – What is this Yanji? A town? Province? A mountain? should be given. Similarly Xanbian, in the title. Readers will not know what are these? A province? A town? Should be given.

lines 46-60 – gD in other mammal herpesviruses responsible for fusion of the cell and viral envelope. Here with FHV-1? Should be given the biological role of the gD molecule.

Mat. Meth.

2.2. This paragraph should be rewritten, as it is not understandable as it stands. What kind of samples you mean? Blood, swabs, tissue? What do you mean by a sample? How many cats were involved? The 93 samples mean one animal, or 93? You screened the samples by FHV-1 PCR and the only positive sample was used in virus isolation? Is it correct? Then write this, or the truth. How did you select cats for the study? Why was 93 the number of the animals? There are no more cats in Yanji? Stray cats, breeder cats, and where from?

line 93 – how long were the swabs kept in saline before processing, and at what temperature? Should be given.

line 95 – Features of the F81 cells? Is it a feline cell line? Which tissue, cell type? Culture conditions? Media serum, %, etc. Why did not the authors tried to isolate virus from all of their 33 PCR positive samples? Alphaherpesviruses from swabs of ongoing, acute infections are easy to isolate. Lot of isolates could have been compared.

line 92 – how did you know that a cat is FHV-1 infected

line 100 – if the virus was isolated at the first passage, why did You passaged it forward six times?

line 104 – Why was it necessary to determine TCID50?

line 136-138 – Gene bank numbers should be given in the result chapter.

Results:

First the authors must give here that how many samples (swabs) were tested and how many of them proved to be by the PCR assay. This must be the first sentence of the Result chapter.

3.3 and 3.4. PCR specifically proved the presence of FHV-1 in the sample(s)? Why the authors used EM and IF? What did these methods add to the previous PCR study? These works seem to me useless.

lines182-183 – should be part of Math. meth.

Figure 5 – I do not see any differences. If there are no differences, why Figure 5 is necessary?

Discussion

Discussion chapter is for analysing, explaining our results.

 244 - 255 – This rows should be part of the introduction chapter, not the discussion.

256-266 – should be part of the result chapter

line 291 – lions? Are lions living in nature in north-east Chima, near Korea and Russia??

Comments on the Quality of English Language

Spelling

line 14 – is causative agent of feline …..

line 15 – pose a threat

line 28-29 – analy-sis

line 34 – genetically relatively homogenous

line 44 – for life- lifelong

line 58 – D,

line 63 – instead of environments – objects

line 64 – after clinical recovery

line 92 – suspected FHV-1-infected cats

line 218 – the strains described in this study…..

line 220 – outside from China ……..

lines 234-238 and Figure 8. Written form of the viruses and their abbreviated forms should be given somewhere together. It is written (lines 232-237) in full,  and later the abbreviations in caption of Figure 8.. Give them together.

line 237 – hotability?

line 268 – age of infection??? time of …

line 268 – compared to ….

line 294 – widespread spread  …..- not nice. repetition of words.

Reviewer 2 Report

Comments and Suggestions for Authors

The manuscript, written by Yang et al. and entitled "The latest prevalence, isolation, and molecular characteristics of Feline Herpesvirus Type 1 in Yanbian, China,"  described the features of FHV-1 isolates, including epidemiological information. Generally, the article is well written, and the sections are well divided. Since the article contains limited information, in my opinion, the study could be published as "communication." My comments are as follows:

1) Abstract: No conclusions or perspectives are described.

2) Some sections of "materials and methods" should be assembled.

3) The figures 1, 2, 3, and 4 are not relevant for the manuscript, and in my opinion, they should be submitted as secondary files. 

4) The authors should provide a table with epidemiological information (prevalence, etc.).

5) Discussion is critical because there is a lack of information regarding the global epidemiological situation (for example, other studies were conducted in the Campania region of Italy by Amoroso et al., in China by Kim et al., etc.), and there is a lot of repetition of the introductory information. I suggest including this information. 

Author Response

I would like to thank the respected reviewers for their constructive comments on our manuscript” The latest prevalence, isolation, and molecular characteristics of Feline Herpesvirus type 1 in Yanbian, China” (vetsci-3165952). We have considered the comments very carefully and have revised the paper accordingly. All changes to the text and figures are shown in red. Thanks to the Reviewer, I believe that this revised paper has been improved considerably. I hope that the corrections are satisfactory.

Comments 1: [Abstract: No conclusions or perspectives are described.]

Response 1: [Epidemiological surveys revealed that 33 of the 93 samples were positive for FHV-1, the gD gene of 33 samples exhibiting low variation, high homology, and no critical amino acid mutation. This study found that the genetic evolution of the FHV-1 gD gene was stable, expanding the molecular epidemiological data on FHV-1 in cats in Yanji City.] We are very grateful for your constructive comments on our manuscript. [We have made efforts to improve the abstract part and rewrite the conclusions of the abstract part. – page 1, Abstract, and line 23/24,36-38]

Comments 2: [Some sections of "materials and methods" should be assembled]

Response 2: Thank you to the reviewers for pointing out the problems, we think this can more clearly describe the process, and in accordance with the recommendations to update the text. [ We deleted the 2.1 section and modified the new 2.1, 2.2, and 2.3 sections to improve the materials and methods. – page 3, paragraph, and line 94-137]

Comments 3: [The figures 1, 2, 3, and 4 are not relevant for the manuscript, and in my opinion, they should be submitted as secondary files.]

Response 3: Thank you for allowing us to further clarify this important point. [This is a necessary part to confirm that the isolated strain is FHV-1 and there is no other virus interference. To make the article more logical, we have updated the results section.]

Comments 4: [The authors should provide a table with epidemiological information (prevalence, etc.).]

Response 4: Thanks for your careful comments. [In the revised manuscript, your reasonable suggestions have been added and can be found on page 5, paragraph 3.2, and line 187.]

Comments 5: [Discussion is critical because there is a lack of information regarding the global epidemiological situation (for example, other studies were conducted in the Campania region of Italy by Amoroso et al., in China by Kim et al., etc.), and there is a lot of repetition of the introductory information. I suggest including this information. ]

Response 5: [In recent years, an epidemiological survey of the upper respiratory tract of cats in Canada showed that the detection rate of FHV-1 was 29.3% [34]. Epidemiological evaluation of the upper respiratory tract of cats with FHV-1 in Spain revealed that the detection rate of FHV-1 was 28.3% [35]. A retrospective molecular study of feline viral diseases in southern Italy (Campania) showed that the detection rate of FHV-1 was 9.05% [36]. The molecular epidemiological report of a natural infection of feline herpesvirus in Kunshan, southern China, showed that the detection rate of FHV-1 was 21.5% [37]. In the molecular epidemiological investigation of FHV-1 in 16 cities in China, the results showed that the detection rate of FHV-1 was 16.3%, mainly in winter and spring, and the positive rate of FHV-1 in northern China was higher than that in southern cats [38]. In this study, the Yanbian area was in the north of China, and the positive detection rate of the FHV-1 epidemiological investigation was 32.2%, with an increasing infection rate compared to other regions. This indicates the high prevalence of FHV-1 in the Yanbian region and it is possible lack of efficacy of the FHV-1 vaccine, reminding the urgency of developing an effective vaccine.] [The authors agree with the reviewer that the article lacks the global prevalence of FHV-1. As suggested by the reviewer, the authors have researched and added more literature to support this. The discussion section was deleted to reduce the repetitive introduction information.– page 10, paragraph 4, and lines 241-295]

Reviewer 3 Report

Comments and Suggestions for Authors

Simple summary is too short and should be extended to cover at least two thirds of the abstract

L26 the diameter of the strain or of the virus?

L28-29 sentence should be improved, since absent of crucial mutations is not phylogenetic results

the authors make a systematic error, i.e. they do not leave a space between the text and the square brackets used for citations. It should be …genetically [1]., not …genetically[1].

L50 from the sentence it is not entirely clear if it refers to whole genome of the virus, or to TR and IR

L57-58 why not: The third ORF encodes gD, which is the…

L66 “and there are individual differences in the protective effect of the vaccine on cats [17].” does it depend on the breed of the cat, the housing condition, feeding, etc.?

L20-21 authors in the abstract says that “and the spread of this disease affects the protection of rare feline animals”; however, there is no information in INTRODUCTION section on the FHV-1 in wild representatives of the Felidae family, species, geographic regions, etc., authors should add some of this data in INTRODUCTION

L72 for non Chinese readers please add what is Yanji, city, region, province (also add this in abstract)

L79 is there the same 93 samples? More information is needed here, what diseases did the cats have? Information on their age, breeds also might be important

L86-87 the meaning of the sentence is not clear, please rewrite. For what other viruses’ cats were tested?

L109 what reaction, PCR?

L114-118 the procedure is not enough described, how samples were prepared, what the procedure was, what materials were used, what microscope was used, in which laboratory the procedure was performed…?

“A total of 93 clinical samples suspected of having an FHV-1 infection 133 were collected from animal hospitals in Yanji. The clinical symptoms included fever, 134 runny nose, conjunctivitis, and other symptoms” this sentence does not belong to 2.6

L138-141 it should be moved to 2.7

L145 please add accessed on  data month 2024 (for instance (http://www.ncbi.nlm.nih.gov, accessed on  30 July 2024)

L147-148 incorrect, change to using bootstrap test with 1000 replications.

L148 “The amino acid sequence was compared with the vaccine strain” it is not clear, add more information, how it was compared, where the sequence of the vaccine strain can be found, etc.

Tittle styles in 2.1, 2.2 and 2.3 are correct, check others, for instance 3.1 is not correct

L157 “brilliant” word is here not appropriate here

L158 what is “FPV, FCV, and FCoV” full names not abbreviations also should be added

L161 in methods it was not described what was used as negative control

L162 it is description of cause-effect relationship but not the title

L164 Under light microscope?

L168 Figure caption must be below the figure but not above

L168 the name of the figure should be rewritten

Figure 2 scale bar should be prepared using the more rounded numbers 100, 500, 1000, but not 300

L170-173 I do not see link between two sentences, …which was like that of other herpesviruses and therefore FHV-1 strain was identified as YBYJ-1. There needs to be a clearer explanation of why the YBYJ-1 strain was identified, it is not clear at the moment

L179-180 please fix format, caption of the figure, as well as scale bar. Now it is very messy

L181 the title is unsuitable

L182-184 what was percentage detection rate?

L195 what du you mean by incomplete vaccination? How many vaccination round are needed for the whole vaccination? Such details are absent also in INTRODUCTION

L184 sequences not sequence

In table 1 what do you mean by “(mon)”?

L198-199 correct this “GenBank database. (Figure 5).”

L200 what do you mean by “no significant difference”? so was there difference or not?

L197 is there need for the special program to see what there is no difference, why not use MEGA software for the same purpose?

L203-205 repetition of what was said in METHODS

L207-208 I do not understand the essence of this sentence

L214 caption of the figure must be after the figure

Figure 6 does not show some very important data, as is clear from the description of the text, it should be deleted or moved to supplementary material

Results of the phylogenetic data should be descried in the separate subsection 3.7

215-242 the presentation of phylogenetic data is not suitable for the publication. Phylogenetic trees should be traditional rectangular not circle. Authors cannot speak of phylogenetic groups, their distinction is unsupported, not significant. There are no bootstrap  support values next to branches, soi it seems that authors do not know how to present and interpret phylogenetic results.

The discussion is not complete, not divided into subsections

L249 correct style FHV-1. FHV-1

L254-255 the study lacks relevancy and novelty “This is the first report on the epidemiological investigation of FHV-1 in Yanji” so authors can repeat study in many other cities and it would be enough for the publication of multiple articles

CONCLUSIONS should be expanded now only three lines.

References do not meet requirements of MDPI journals

 Simple summary is too short and should be extended to cover at least two thirds of the abstract

L26 the diameter of the strain or of the virus?

L28-29 sentence should be improved, since absent of crucial mutations is not phylogenetic results

the authors make a systematic error, i.e. they do not leave a space between the text and the square brackets used for citations. It should be …genetically [1]., not …genetically[1].

L50 from the sentence it is not entirely clear if it refers to whole genome of the virus, or to TR and IR

L57-58 why not: The third ORF encodes gD, which is the…

L66 “and there are individual differences in the protective effect of the vaccine on cats [17].” does it depend on the breed of the cat, the housing condition, feeding, etc.?

L20-21 authors in the abstract says that “and the spread of this disease affects the protection of rare feline animals”; however, there is no information in INTRODUCTION section on the FHV-1 in wild representatives of the Felidae family, species, geographic regions, etc., authors should add some of this data in INTRODUCTION

L72 for non Chinese readers please add what is Yanji, city, region, province (also add this in abstract)

L79 is there the same 93 samples? More information is needed here, what diseases did the cats have? Information on their age, breeds also might be important

L86-87 the meaning of the sentence is not clear, please rewrite. For what other viruses’ cats were tested?

L109 what reaction, PCR?

L114-118 the procedure is not enough described, how samples were prepared, what the procedure was, what materials were used, what microscope was used, in which laboratory the procedure was performed…?

“A total of 93 clinical samples suspected of having an FHV-1 infection 133 were collected from animal hospitals in Yanji. The clinical symptoms included fever, 134 runny nose, conjunctivitis, and other symptoms” this sentence does not belong to 2.6

L138-141 it should be moved to 2.7

L145 please add accessed on  data month 2024 (for instance (http://www.ncbi.nlm.nih.gov, accessed on  30 July 2024)

L147-148 incorrect, change to using bootstrap test with 1000 replications.

L148 “The amino acid sequence was compared with the vaccine strain” it is not clear, add more information, how it was compared, where the sequence of the vaccine strain can be found, etc.

Tittle styles in 2.1, 2.2 and 2.3 are correct, check others, for instance 3.1 is not correct

L157 “brilliant” word is here not appropriate here

L158 what is “FPV, FCV, and FCoV” full names not abbreviations also should be added

L161 in methods it was not described what was used as negative control

L162 it is description of cause-effect relationship but not the title

L164 Under light microscope?

L168 Figure caption must be below the figure but not above

L168 the name of the figure should be rewritten

Figure 2 scale bar should be prepared using the more rounded numbers 100, 500, 1000, but not 300

L170-173 I do not see link between two sentences, …which was like that of other herpesviruses and therefore FHV-1 strain was identified as YBYJ-1. There needs to be a clearer explanation of why the YBYJ-1 strain was identified, it is not clear at the moment

L179-180 please fix format, caption of the figure, as well as scale bar. Now it is very messy

L181 the title is unsuitable

L182-184 what was percentage detection rate?

L195 what du you mean by incomplete vaccination? How many vaccination round are needed for the whole vaccination? Such details are absent also in INTRODUCTION

L184 sequences not sequence

In table 1 what do you mean by “(mon)”?

L198-199 correct this “GenBank database. (Figure 5).”

L200 what do you mean by “no significant difference”? so was there difference or not?

L197 is there need for the special program to see what there is no difference, why not use MEGA software for the same purpose?

L203-205 repetition of what was said in METHODS

L207-208 I do not understand the essence of this sentence

L214 caption of the figure must be after the figure

Figure 6 does not show some very important data, as is clear from the description of the text, it should be deleted or moved to supplementary material

Results of the phylogenetic data should be descried in the separate subsection 3.7

215-242 the presentation of phylogenetic data is not suitable for the publication. Phylogenetic trees should be traditional rectangular not circle. Authors cannot speak of phylogenetic groups, their distinction is unsupported, not significant. There are no bootstrap  support values next to branches, soi it seems that authors do not know how to present and interpret phylogenetic results.

The discussion is not complete, not divided into subsections

L249 correct style FHV-1. FHV-1

L254-255 the study lacks relevancy and novelty “This is the first report on the epidemiological investigation of FHV-1 in Yanji” so authors can repeat study in many other cities and it would be enough for the publication of multiple articles

CONCLUSIONS should be expanded now only three lines.

References do not meet requirements of MDPI journals

Comments on the Quality of English Language

Some of the English mistakes are given in my comments to authors, but in general the style of the sentences is not good and should be carefully reviewed by an English language editor by proofreading service

Reviewer 4 Report

Comments and Suggestions for Authors

This paper is basically a survey of the prevalence of feline herpesvirus 1 (FHV-1) in cats with upper respiratory tract (URT) infection in Yangi, China. FHV-1 is one of the two main causes of URT disease in cats, and this study reports that 33 (32.2%) of 93 samples from cats with URT disease were positive for FHV-1, although the clinical criteria for selection of cases is not made clear in the methods. Although vaccination protects reasonably well against disease, it does not prevent (though may reduce) shedding after infection. The virus becomes latent in trigeminal ganglia after primary infection, and may reactivate – as with many other herpesviruses - particularly following stress which can also make control difficult.

The authors cover a great deal of background in the Introduction, and although a brave attempt to survey the literature has been made, it could be shortened and made more relevant to this particular study, ie a descriptive epidemiological prevalence study. There are also some errors in interpretation, some which may be due to incorrect English. A list of these is below.

The Materials and Methods are presented in considerable detail, where normally some of this eg cell and virus culture would be referenced to earlier papers. But it is appreciated that this area of study may be new to the authors of this paper, and so is probably acceptable to some extent. There should also be more detail about hospitals selected and the clinical criteria for selection of cats.

The Results section is rather muddled. It would be helpful to start with the findings of the main purpose of the study, ie a survey of FHV-1 prevalence in this area of China. This is the proportion of the samples that were identified as FHV-1 positive, as in the relevant sections xxx below. The main characteristics (eg number of cats from each hospital, clinical signs, vaccine status and age) of both the positive (table 1) and negative cats should be summarised briefly and compared (may need a table for the negative cats as well).

The next results sections should be in the order:

·         Preliminary identification should be cell culture findings ie characteristic FHV-1 cytopathic effect (and reference) and remove unnecessary detail about 50% TCIDs etc.  A comment about feline calicivirus which is also a common cause of URT infection should also be made. The CPE is different, but can sometimes be muddled.

·         EM and immunofluorescence: make it clear which strains (just YBYJ-1?) were subjected to this. The EM photo is not very clear and it may not be worth including, since there is other confirmation that the isolate(s?) is FHV-1, ie IF and sequencing.

·         gD gene sequence data (which confirms all the isolates are FHV-1) and analyses.

The Discussion starts with a review of the FHV-1 literature, which is rather too detailed and has some inaccuracies in it (see list below). The first focus should be a brief resume of why the study was carried out; the results of the survey; and to compare and contrast the findings with other relevant studies as in done in a later paragraph.

This can be followed by a brief discussion of how the viruses were identified and confirmed by characteristic CPE (ref), IF (in some/one case?), but maybe not EM as photos not very clear. No detail on 50% TCID detail etc is needed. This could be followed by discussion of the gD sequence analyses including the comparative phylogeny.

The final part of the discussion could be putting the results in the broader context of the Yanbian region as the authors have done.

List of specific points.

Abstract.

Line 19. The statement should read ‘Feline herpesvirus type 1 (FHV-1), also known as feline viral rhinotracheitis (FVR) virus, is one of the main causes of URT disease in cats.’

Lines 26/27. No need to mention strain diameter and 50% TCID. Need to make it clear in the abstract ioslates confirmed as FHV-1 by characteristic CPE and gD sequence analysis.

Lines 27/28 Survey results should be towards the start of the Abstract and state where the samples were from ie cats with URT disease, and from hospitals.

Introduction.

Line 38. Corneal ulcers may occur but are rare in FHV-1 infection. Reference 3 is not appropriate here for the main list of clinical signs.

Line 42. Are refs 5 and 6 correct here?

Line 44. Latent infection does not ‘seriously affect its health’. Recrudescence in recovered cats generally only occurs rarely, and clinical signs are typically mild.

Line 54. ‘is’ not ‘are’

Line 64/65. ‘….can persist after a cure, and it is difficult to completely eradicate the  virus’. This is a bit muddled and inaccurate. The virus goes latent in the trigeminal ganglia in virtually all infected , but clinically recovered cats, and in this sense it is difficult to eradicate on a population basis.

Line 71 – evidence for this statement about vaccine effectiveness failing?

Materials and Methods.  See also comments above.

Line 79 – how many cats and how many hospitals? [Also at some point need to comment on why nasal swabs cf to more usual oro-pharyngeal swabs? Sometimes difficult to get enough discharge on nasal swabs.]

Line 84 ‘were collected…’ need to add ‘for cats with URT disease’.

Line 92 suspected FHV-1 infected cats.

Line 100. Why passage 6 times? Could understand if plaque/or end dilution purifying.

Line 121 change ‘group’ to culture flask.

Lines 133-135 completely in the wrong place. See above comments.

Line 148 – which vaccine virus?

Results. See above for main comments.

Line 164.  should be ‘grape-like’ clusters.

Line 166/167 why the need for this detail?

Lines 183/184. See above re re-location of this info.

Line 237 –‘notability’ – sense unclear - is this a typo?

Line 239 Figure 8. The abbreviations of viruses names should bracketed after full names in lines 232-235.

Discussion. Main points are above.

Lines 244/245. Main clinical signs of conjunctivitis, ocular and nasal discharges, sneezing and sometimes coughing are missing. Chronic gingivostomatitis is much more likely to be associated with feline calicivirus.

Line 252 ‘often causes weakening’ – not sure what this means.

Line 270 – is reference 30 the right one here- it refers to cheetahs

Line 272 – reference to breed of cat in this study? If so, needs to be mentioned in the clinical results, as outlined above.

Line 274 – any info on housing for cats in this study?

Line 286 – serological efficacy of FHV-1? Should perhaps be ‘possible lack of efficacy of the FHV-1 vaccine’?

Comments on the Quality of English Language

generally fine, but some use of words needs correcting , as in report.

Round 2

Reviewer 1 Report

Comments and Suggestions for Authors I have read the authors replies to my comments, I agree, accept their replies, and I suggest this paper for publication in its present form. Comments on the Quality of English Language I have read the authors replies to my comments, I agree, accept their replies, and I suggest this paper for publication in its present form.

Reviewer 2 Report

Comments and Suggestions for Authors

The authors have improved the quality of the manuscript following the reviewers' suggestions. In my opinion the article is acceptable.